# Thalamic nucleus reuniens coordinates prefrontal-hippocampal synchrony to suppress extinguished fear

Michael S. Totty [1,2], Tuğçe Tuna [1,2], Karthik R. Ramanathan[1,2], Jingji Jin [1,2], Shaun E. Peters[1] & Stephen Maren [1,2] ✉

Traumatic events result in vivid and enduring fear memories. Suppressing the retrieval of these memories is central to behavioral therapies for pathological fear. The medial prefrontal cortex (mPFC) and hippocampus (HPC) have been implicated in retrieval suppression, but how mPFC-HPC activity is coordinated during extinction retrieval is unclear. Here we show that after extinction training, coherent theta oscillations (6−9 Hz) in the HPC and mPFC are correlated with the suppression of conditioned freezing in male and female rats. Inactivation of the nucleus reuniens (RE), a thalamic hub interconnecting the mPFC and HPC, reduces extinction-related Fos expression in both the mPFC and HPC, dampens mPFC-HPC theta coherence, and impairs extinction retrieval. Conversely, theta-paced optogenetic stimulation of RE augments fear suppression and reduces relapse of extinguished fear. Collectively, these results demonstrate a role for RE in coordinating mPFC-HPC interactions to suppress fear memories after extinction.

For many individuals, traumatic events result in vivid and enduring memories that drive unrelenting and pathological fear[1]. Behavioral interventions, such as prolonged exposure therapy, rely upon extinction learning to suppress emotional memories and dampen fear responses[2,3]. Considerable work demonstrates that the mPFC is critical for extinction learning and may be an essential neural substrate for the suppression of episodic fear memories after extinction[4–7]. Work in both humans and animals suggests that interactions between the mPFC and HPC are critical for the suppression of episodic memories[8,9]. Indeed, inhibiting the retrieval of context-inappropriate memories, a hallmark of retrieval suppression, is critical for the expression of context-dependent extinction[10,11]. Hence, mPFC-HPC interactions may prevent intrusive fear memories from disrupting extinction retrieval[12–14], though the mechanisms supporting this function are unknown.

Neural oscillations facilitate interactions between brain regions by coordinating and synchronizing neural activity[15]. Investigations into the neural oscillations underlying aversive learning and memory[16] have discovered that theta-range oscillations act to synchronize limbic structures during the retrieval of both fear[16–21] and extinction

memories[18,22,23]. For example, considerable evidence implicates synchronization of prefrontal-hippocampal oscillations in memory retrieval[24–28]. However, it is unclear how the mPFC influences the retrieval of hippocampus-dependent fear and extinction memories insofar as the mPFC has only sparse projections to the HPC[29,30]. One candidate brain region interconnecting the mPFC and HPC is the thalamic nucleus reuniens (RE). The RE is a small structure in the ventral midline thalamus that bidirectionally connects the mPFC and HPC[31] and is a critical hub for coordinating mPFC-to-HPC interactions[32]. Indeed, Ramanathan and colleagues found that the RE, and mPFC→RE projections, are necessary for the encoding and retrieval of both auditory and contextual extinction memories[33,34]. Inactivation of the RE also impairs the precision of contextual fear memories[34,35] a process that requires both the HPC and mPFC[36,37]. Additionally, inactivating the RE has been reported to impair mPFC-HPC synchrony and inhibitory projections from the RE to HPC[38] are positioned to mediate retrieval suppression processes implicated in extinction[8,39–41]. Collectively, these data suggest that RE may be critical for mPFC-HPC synchrony during the retrieval of extinction memories.

[1]Department of Psychological and Brain Sciences, Texas A&M University, College Station, TX, USA. [2]Institute for Neuroscience, Texas A&M University, College Station, TX, USA. ✉e-mail: maren@tamu.edu

Here we test this specific hypothesis using pharmacological, optogenetic, and electrophysiological methods in behaving rats. We found that the mPFC and RE display distinct 3–6 Hz and 6–9 Hz oscillations that are positively and negatively, respectively, correlated to freezing behavior during extinction acquisition and retrieval. Extinction retrieval was associated with increased theta coherence between the mPFC and HPC. Pharmacological inactivation of the RE induced fear relapse after extinction, impaired c-Fos immediate early gene expression in the mPFC and HPC, and abolished mPFC-HPC theta synchrony. Control conditions employing non-conditioned rats demonstrate that increases in freezing behavior induced by RE inactivation are memory-specific. Finally, we show that theta-paced optogenetic stimulation of the RE is sufficient to block fear relapse. These data demonstrate that RE is a critical hub for the retrieval of safe extinction memories and may be a unique therapeutic target for the suppression of intrusive traumatic memories.

## Results

### Fear extinction is characterized by mPFC-HPC theta coherence

To examine the role of mPFC-HPC theta synchronization in fear extinction, we simultaneously recorded local field potentials (LFPs) (Fig. 1A) from both the prelimbic (PL) and infralimbic (IL) regions of the mPFC and the CA1 region of the dorsal HPC (Fig. 1C, D) during extinction acquisition and retrieval. Rats ($n = 6$; 3 male and 3 female) were conditioned using a standard auditory fear conditioning procedure (Fig. 1B). After conditioning, recordings were made during the context exposure, fear extinction, and extinction recall sessions.

As shown in Fig. 1E, all animals acquired fear of the CS, which was indicated by increased freezing across conditioning trials (main effect of trials: $F_{5,20} = 5.44$, $p = 0.003$). The following day, all rats were exposed to the extinction context for 20 min where they displayed low levels of freezing. During extinction training, all rats showed high fear of the CS presentations during the early extinction trials and freezing decreased during successive CS presentations (main effect of trials: $F_{9,36} = 6.053$, $p < 0.0001$). CS-elicited freezing behavior remained low during retrieval testing the following day, demonstrating a robust extinction memory.

We chose to focus our electrophysiological analyses on three sessions in which animals showed different levels of conditioned fear: context exposure ('Expo', low fear), high fear (first five trials of extinction session, 'Fear'), and suppressed fear (first five trials of extinction retrieval, 'Ext'). To control for differences in overall spectral power between animals (while also ignoring the influence of higher frequency bands) the oscillatory power for all recordings was normalized to the total power from 2 to 12 Hz. As shown in Fig. 1F, conditioned freezing during these sessions was significantly higher during early extinction ('Fear') relative to the exposure ('Expo') and extinction retrieval ('Ext') sessions (main effect of Days: $F_{2,10} = 9.664$, $p = 0.006$). Examining the LFP recordings, we found that the expression of conditioned freezing during the earliest extinction trials ('Fear') was associated with greater 3–6 Hz power (main effect of Day: $F_{2,30} = 46.53$, $p < 0.0001$) in the IL (Fear vs Ext post hoc: $p < 0.0001$), PL (Fear vs Ext post hoc: $p < 0.0001$), and HPC (Fear vs Ext post hoc: $p = 0.0062$) relative to the extinction retrieval session (Fig. 1G–M). In contrast, the suppression of conditioned freezing during extinction retrieval ('Ext') was associated with greater 6–9 Hz power (main effect of Day: $F_{2,30} = 13.89$, $p < 0.0001$) in the IL (Fear vs Ext post hoc: $p = 0.002$) and PL (Fear vs Ext post hoc: $p = 0.0092$), but not the HPC (Fear vs Ext post hoc: $p = 0.128$) relative to the early extinction trials. Moreover, we found that, across all sessions, 3–6 Hz power was positively correlated with conditioned freezing in both IL ($r^2 = 0.406$, $p = 0.004$; Fig. 1J) and PL ($r^2 = 0.235$, $p = 0.042$; not shown), whereas freezing behavior was negatively correlated with 6–9 Hz power in the IL ($r^2 = 0.401$, $p = 0.005$) (Fig. 1M). The correlation between freezing behavior and 6–9 Hz oscillations in the PL was also negative, but was not statistically

significant ($r^2 = 0.207$, $p = 0.058$). These results further demonstrate that freezing behavior is associated with an increase in ~4 Hz oscillatory power in the mPFC.

Next, we sought to determine if the retrieval of extinction memories was associated with increased mPFC-HPC theta synchrony (Fig. 1N). Considering that 6–9 Hz oscillations are negatively correlated with conditioned freezing (e.g., Fig. 1M), we predicted that fear extinction would also be associated with mPFC-HPC coherence in this frequency band. Indeed, we found that mPFC-HPC coherence is primarily centered at ~8 Hz (Fig. 1O). The HPC showed increased coherence (6–9 Hz) with the IL (main effect: $F_{2,10} = 16.69$, $p = 0.0007$; Expo vs Ext post hoc: $p = 0.0006$) and PL (main effect: $F_{2,10} = 13.94$, $p = 0.0013$; Expo vs Ext post hoc: $p = 0.001$) during extinction retrieval compared to context exposure (Fig. 1P). Importantly, these results cannot be explained by differences in freezing behavior as neither 3–6 nor 6–9 Hz coherence is correlated with freezing behavior across sessions (SFig. 1). This demonstrates that the retrieval of extinction memories is associated with 6–9 Hz theta synchrony between the mPFC and HPC.

### The thalamic nucleus reuniens exhibits theta-range oscillations during extinction that correlate with conditioned freezing

We next sought to determine if the RE displays similar oscillations to the PFC and HPC during fear extinction (Fig. 2A). To this end, a multielectrode array was implanted in the RE one week prior to behavioral testing; bipolar LFP recordings were conducted by referencing one channel to another intra-RE channel to remove volume conducted signals from external sources (Fig. 2B). All rats ($n = 4$; 2 male and 2 female) were fear conditioned as previously described.

During fear extinction, rats exhibited high levels of conditioned freezing behavior that extinguished over 45 CS-alone trials (Fig. 2C, D). Like the previous experiment, we chose to restrict our electrophysiological analyses to the first and last 5 trials of extinction to examine differences in high and low fear states, respectively. Similar to the mPFC and HPC, we observed prominent 3–6 Hz oscillations (Fig. 2E) during the first five trials of extinction (Early Ext), and this switched to 6–9 Hz oscillations (Fig. 2F) during the last five trials (Late Ext). This was confirmed by focusing on the relative spectral power during the CS (Fig. 2G), showing that 3–6 Hz power decreased, whereas 6–9 Hz power increased, from Early to Late Ext (Frequency x Block interaction: $F_{1,6} = 7.376$, $p = 0.035$). We further show that 3–6 Hz rhythms positively correlated with freezing behavior ($F_{1,6} = 7.233$, $r^2 = 0.547$, $p = 0.036$). These data confirm that the RE also exhibits 4 and 8 Hz rhythms dependent on behavioral state.

### Fear relapse induced by RE inactivation impairs c-Fos expression in the mPFC and HPC

Previous work found that RE is critical for the retrieval of extinction memories[33], and we hypothesize that the RE mediates this process by driving neural activity in the mPFC and HPC. To test this hypothesis, we first determined if fear relapse induced by RE inactivation impaired immediate-early gene (c-Fos) expression, a proxy for neural activity, in the mPFC and HPC. To this end, rats were implanted with a single midline cannula targeting the RE. One week after surgery, all rats underwent standard fear conditioning and extinction. Conditioning and extinction behavior were like that observed in the previous experiments and there were no differences between drug groups ($p > 0.3$). As shown in Fig. 3A, rats that were infused with the GABA$_A$ agonist, muscimol (MUS; $n = 14$ males) exhibited poor extinction recall compared to saline (SAL; $n = 14$ males) controls (main effect of Drug: $F_{1,27} = 8.66$, $p = 0.007$). All rats were then sacrificed 90 min after retrieval testing to examine Fos expression. We found that MUS animals displayed decreased Fos expression, an indication of reduced neural activity, in the HPC (Fig. 3C; $t_{27} = 3.26$, $p = 0.003$) as well as both PL ($t_{27} = 2.59$, $p = 0.015$) and IL ($t_{27} = 2.45$ $p = 0.021$) subregions of the

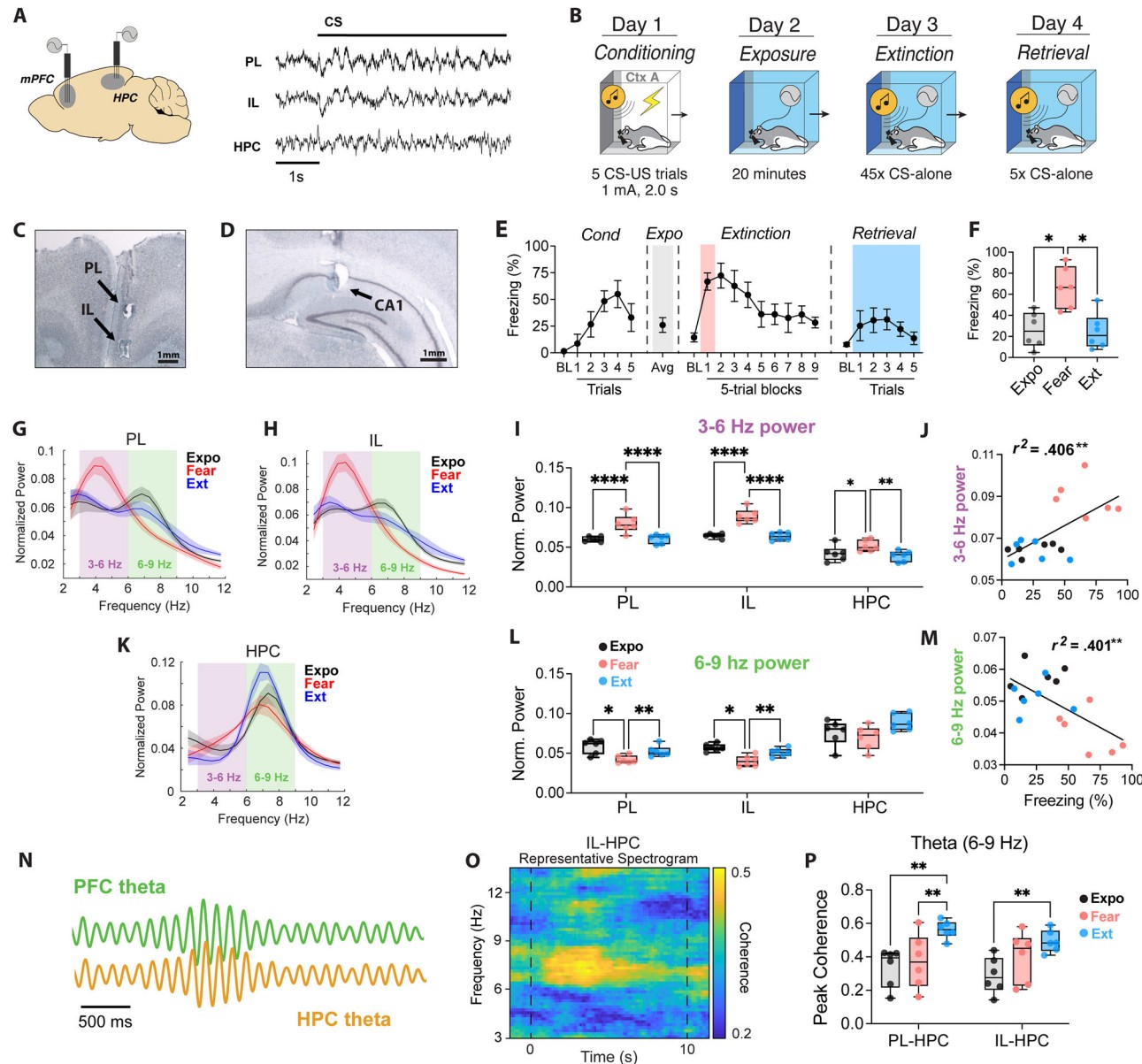

**Fig. 1 | In vivo electrophysiological recordings during context exposure, fear extinction, and extinction retrieval. A** Illustration and representative traces of mPFC and HPC local field potential recordings (LFPs). **B** Behavioral timeline of the fear conditioning protocol. Representative photomicrographs show mPFC (**C**) and HPC (**D**) electrode placements. Similar photomicrographs were taken to confirm electrode placement for all animals. **E** Averaged freezing behavior ($n = 6$; 3 male and 3 female) during fear conditioning, context exposure, extinction, and extinction retrieval (BL = baseline). **F** Average freezing during fear retrieval during the first five trials of extinction (Fear) was greater than context exposure (Expo; $p = 0.044$, repeated measures one-way ANOVA with Tukey's multiple comparison test) and the five extinction retrieval trials (Ext; $p = 0.024$). **G, H, K** Normalized spectral power of theta-range oscillations in the prelimbic (PL) and infralimbic (IL) cortex and the HPC during 10 s CS presentations across days. **I** Normalized 3–6 Hz power was elevated during Fear recall in the PL relative to Exposure ($p < 0.0001$) and Extinction

($p < 0.0001$). This was also observed in the IL ($p < 0.0001$; $p < 0.0001$) and HPC ($p = 0.048$; $p = 0.006$). **L** Conversely, 6–9 Hz power was decreased during Fear recall in the PL relative to Exposure ($p = 0.0305$) and Extinction ($p < 0.002$). This was likewise observed in the IL ($p = 0.0109$; $p = 0.0092$). **J, M** Linear regression of freezing vs IL theta-range power during Expo, Fear and Ext. **N** Cartoon illustrating synchronous theta oscillations. **O** Example coherogram depicting high mPFC-HPC theta coherence during extinction retrieval. **P** Average peak coherence of 6–9 Hz oscillations between the PL and HPC was highest during extinction retrieval, compared to exposure ($p = 0.002$) and fear retrieval ($p = 0.008$). IL-HPC coherence was also greater during extinction retrieval relative to context exposure ($p = 0.005$). BL baseline, CS conditioned stimulus, PL prelimbic cortex, IL infralimbic cortex, HPC hippocampus. Line plots represent mean ± s.e.m.s; boxplots represent mean plus minima and maxima with lower and upper quantiles. *$p < 0.05$; **$p < 0.01$; ***$p < 0.001$; ****$p < 0.0001$. Source data are provided as a Source Data file.

mPFC (Fig. 3D), compared to SAL controls. As a control for non-specific effects we additionally quantified Fos expression in the medial geniculate nucleus (MGN), a thalamic region implicated in auditory fear conditioning with no direct connections to the RE. We found that MGN Fos was not affected by RE inactivation ($t_{27} = 0.71$, $p = 0.485$). This suggests that RE inactivation results in impaired neuronal activity in both the PFC and HPC.

## Pharmacological inactivation of RE impairs both mPFC-HPC synchrony and extinction retrieval

It has previously been shown that RE is critical for both the retrieval of extinction memories[33,34] and mPFC-HPC synchrony during a working memory task[39]. However, it is unknown if inactivating the RE impairs mPFC-HPC synchrony associated with fear extinction. To test this, we implanted electrodes in the mPFC and HPC and a cannula in the RE to

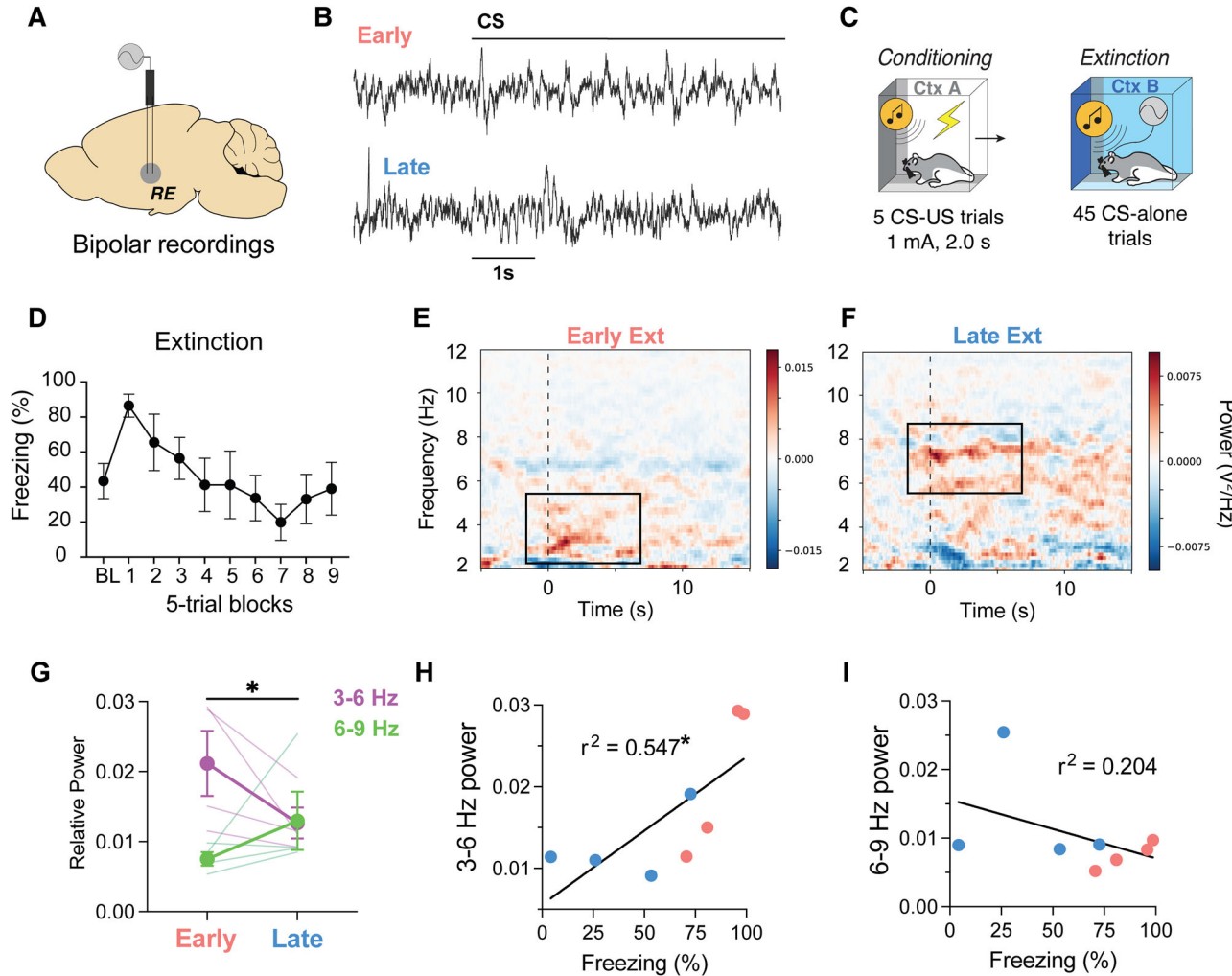

**Fig. 2 | The nucleus reuniens displays theta-range rhythms that correlate with conditioned freezing across extinction learning. A** Illustration and (**B**) raw traces of bipolar electrophysiological field recordings from the RE. **C** Behavioral timeline showing that field recordings took place during fear extinction after all rats were conditioned the previous day in distinct contexts. **D** Average freezing behavior (*n* = 4; 2 male and 2 female) during fear extinction training. Baseline subtracted spectrograms showing increased 3–6 Hz power (**E**) during the first five trials of extinction (Early) and increased 6–9 Hz power (**F**) during the last five trials of extinction (Late). **G** Average normalized power for 3–6 Hz oscillations decreased from early to late extinction, whereas 6–9 Hz power increased (*p* = 0.041, two-way ANOVA). Relationship of freezing behavior vs 3–6 Hz power (**H**) and 6–9 Hz power (**I**). RE = nucleus reuniens of the thalamus, CS conditioned stimulus. Line plots represent mean ± s.e.m.s. *\*p* < 0.05. Source data are provided as a Source Data file.

record mPFC and HPC LFPs while pharmacologically inactivating the RE (Fig. 4A). One week after surgery, all animals (*n* = 11; 8 male and 3 female) underwent standard fear conditioning and extinction as previously described (Fig. 4B). For extinction retrieval testing, rats were infused with either muscimol (MUS) or saline (SAL) immediately prior to testing using a within-subjects design (drug infusions were counterbalanced for test order).

As shown in Fig. 4D, all animals acquired conditioned freezing to the CS (main effect of trials: $F_{5,45}$ = 21.30, *p* < 0.0001) and subsequently extinguished freezing to the CS (main effect of blocks: $F_{9,72}$ = 3.041, *p* = 0.0039). Like the previous experiment, we found that muscimol inactivation of the RE induced fear memory relapse during retrieval testing. This was manifest by increased freezing behavior to CS presentations (Trials x Drug interaction: *F* = 4.369, *p* = 0.0037). To determine if fear relapse induced by RE inactivation is associated with decreased mPFC-HPC synchrony, theta-range coherence was averaged across the five retrieval trials for each animal and the peak coherence value in 6–9 Hz range was found (Fig. 4F). Comparing these peak coherence values, we found that muscimol inactivation of the RE decreased IL-HPC (muscimol vs saline post hoc: *p* = 0.017) 6–9 Hz theta

coherence (main effect of drug: $F_{1,20,}$ = 6.736, *p* = 0.017). Coherence in the lower 3–6 Hz range was unaffected (SFig. 2). Collectively, these results demonstrate that the RE is indeed critical to both the retrieval of extinction memories and its associated mPFC-HPC theta synchrony.

**Optogenetic inhibition of RE selectively impairs extinction memory retrieval**

Pharmacological inhibition of the RE may have disrupted performance during retrieval testing either by impairing contextual processing prior to CS onset or by affecting extinction memory retrieval during presentation of the extinguished CS. To assess whether RE is involved in extinction retrieval, we adopted an optogenetic strategy to silence the RE immediately prior to the first CS delivery, leaving the RE functionally intact during the pre-CS baseline period (when animals sample the context). We also included a non-conditioned control to determine whether increases in freezing associated with RE inactivation are due to non-specific increases in anxiety[42]. To this end, rats were first injected with an adeno-associated virus encoding with a red-light activated, inhibitory opsin (AAV8-CaMKII-Jaws-GFP) or a control fluorescent protein (AAV8-CaMKII-GFP) into the RE, along with an

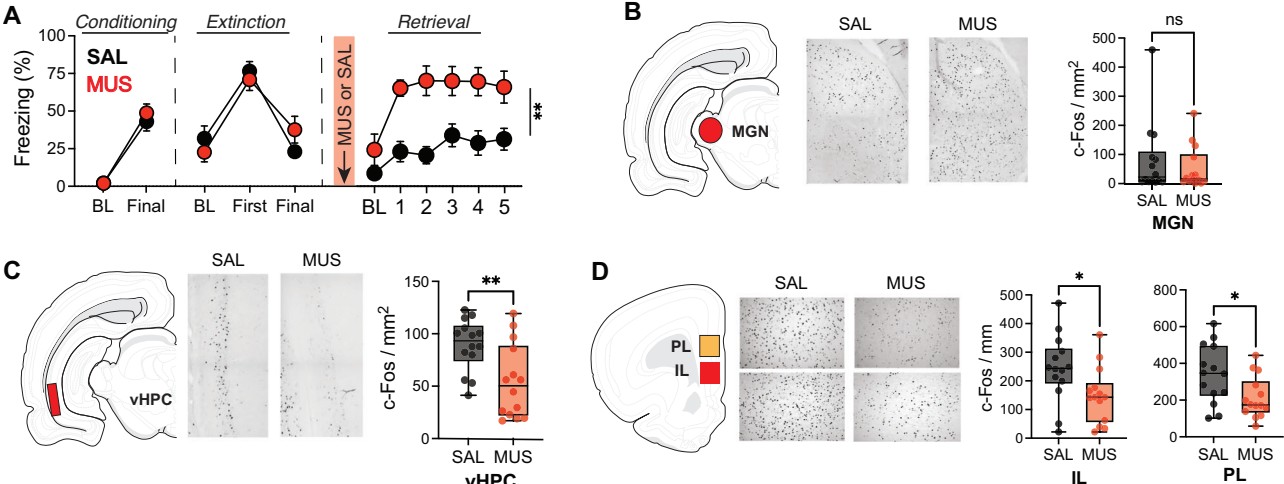

**Fig. 3 | Pharmacological inactivation of RE suppresses Fos expression in mPFC and HPC during after relapse of extinguished fear. A** Freezing behavior during conditioning, first and last trial of extinction, and extinction retrieval after saline (SAL; $n = 14$ males) or muscimol (MUS; $n = 14$ males) infusions into the RE. Fear relapse induced by MUS inactivation of the RE did not impair c-Fos expression in the MGN ($p = 0.485$, unpaired $t$-test) (**B**), but did reduce c-Fos expression in both the vHPC ($p = 0.006$) (**C**) and both IL ($p = 0.031$) and PL ($p = 0.022$) subregions of the mPFC (**D**). MGN medial geniculate nucleus of the thalamus, vHPC ventral hippocampus, IL infralimbic cortex, PL prelimbic cortex. Line plots represent mean ± s.e.m.s; boxplots represent mean plus minima and maxima with lower and upper quantiles. *$p < 0.05$; **$p < 0.01$. Source data are provided as a Source Data file.

optic fiber, four weeks prior to behavioral testing (Fig. 5A). Half of the rats expressing the Jaws virus were conditioned (Jaws; $n = 12$; 5 male and 7 female) while the other half were merely exposed to the CS without US reinforcement to serve as a non-conditioned control group (Control; $n = 8$; 5 male and 3 female). All rats expressing the control GFP virus were conditioned (GFP; $n = 10$; 6 male and 4 female) (Fig. 5B).

During conditioning (Fig. 5C), both Jaws and GFP animals acquired high levels of conditioned freezing, whereas the non-conditioned Control group exhibited low levels of freezing behavior (Trial x Group interaction: $F_{10, 135} = 9.067$, $p < 0.0001$). The following two days, all rats were first extinguished to the conditioning context (Fig. 5D; Block x Group interaction: $F_{16, 216} = 5.024$, $p < 0.0001$) and then to the auditory CS in a novel context (Fig. 5E; Block x Group interaction: $F_{18, 234} = 3.629$, $p < 0.0001$). For retrieval testing, all rats were tested for extinction memory either with (ON) or without (OFF) constant red laser stimulation using a counterbalanced, within-subjects design. If RE inactivation produces an anxiogenic state, one would expect both the Jaws and non-conditioned Control groups to show increased fear during retrieval testing. However, comparing the average freezing during the trial period during laser ON vs OFF conditions (main effect of Laser: $F_{1, 27} = 4.776$, $p = 0.038$; Laser x Group interaction: $F_{2, 27} = 3.374$, $p = 0.049$) (Fig. 5F), we found that optogenetic inactivation of the RE only increased freezing levels in the conditioned Jaws-expressing rats ($p = 0.004$); but not non-conditioned ($p = 0.978$) or GFP-expressing controls ($p = 0.690$). Interestingly, this effect was evident immediately upon optogenetic inhibition prior to the first CS trial (SFig. 3). This effectively demonstrates that freezing induced by RE inactivation is in fact specific to the relapse of extinguished fear, rather than a non-specific anxiogenic effect. Moreover, these results reveal that RE inhibition immediately prior to CS test trials is sufficient to produce extinction retrieval deficits and cause the relapse of extinguished fear.

**Sinusoidal 8 Hz stimulation of the RE prevents fear renewal**
Given that RE inactivation impairs the retrieval of extinction memories, we next aimed to discover if RE stimulation could enhance extinction retrieval and prevent the relapse of extinguished fear that normally occurs when an extinguished CS is presented outside the extinction context (i.e., renewal)[12,43,44]. For this purpose, we chose to

optogenetically stimulate the RE using 8-Hz sinusoidal stimulation, which we hypothesized would mimic the oscillatory synchrony in the mPFC-HPC circuit that is associated with extinction retrieval. To test this, rats were injected with an AAV encoding either an excitatory, blue light-activated opsin (AAV8-CaMKII-ChR2-mCherry) or a control fluorescent protein (AAV8-CaMKII-mCherry), along with either an optic fiber or 4-channel optrode, targeting the RE four weeks prior to behavior (Fig. 6A). Because the RE is critical for mPFC-HPC theta (~6–9 Hz) synchrony, we chose to use an 8-Hz sinusoidal stimulation pattern during each CS presentation (laser onset 5-sec pre-CS and offset 5-sec post-CS) (Fig. 6B). The ChR2 and mCherry-expressing rats ($n = 13$; 5 male and 8 female) that were implanted with optrodes underwent a naive stimulation protocol (see "Methods" section) to validate that sinusoidal stimulation entrains RE LFPs and single-units. They and all other animals then underwent standard fear conditioning and extinction as previously described. Unlike previous experiments, retrieval testing in this experiment took place in a novel context to test for fear renewal (Fig. 6C).

To validate our sinusoidal stimulation protocol, we recorded LFPs and single units while stimulating at 8 Hz for 5 s at a time for 20 trials. Both ChR2 ($n = 5$; 3 male and 2 female) and mCherry-expressing rats ($n = 5$; 2 male and 3 female) were tested under both blue-laser (450 nm) and red-laser (635 nm) conditions (Fig. 6D–H). ChR2 has an excitation range of ~400–550 nm, so it should not be excited by red-laser illumination[45]. Indeed, we found that 8-Hz stimulation strongly induced 8-Hz rhythms (Fig. 6D) only in ChR2-expressing rats under blue-laser stimulation (Fig. 6E–G). Additionally, we found that nearly all (7/8) recorded RE neurons were significantly entrained by 8-Hz stimulation in ChR2-expressing rats. Comparatively, 0/8 of the units recorded in mCherry rats were light-modulated. Of the 7 light-modulated units, the most strongly entrained units [$n = 3$; mean resultant length (MRL) > 0.6] were all entrained to the phase of maximal light power (~90 degrees), whereas other units (MRL 0.04–0.47) were entrained to phases of minimal light power (~225 or 300 degrees) (Fig. 6J and SFig. 4). Interestingly, although 8-Hz stimulation robustly entrained RE units (Fig. 6K), we observed a mixed effect on the firing rates of the units with no significant change in the total average rates (Fig. 6L and SFig. 4). Importantly, 8-Hz stimulation had no effect on locomotion during naive recordings (SFig. 5). Collectively, we show

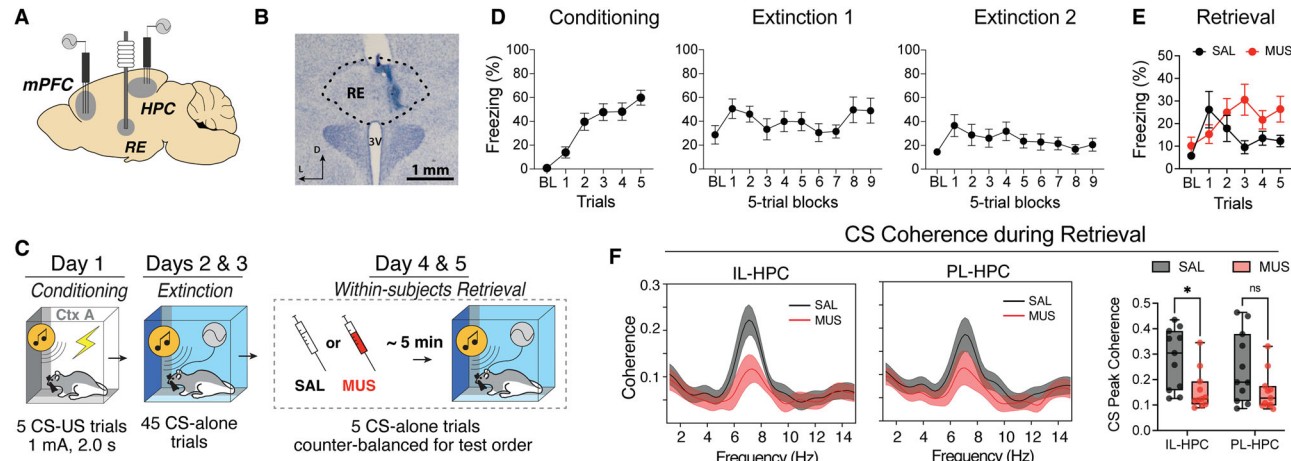

**Fig. 4 | Pharmacological inactivation of RE impairs mPFC-HPC synchrony.**
**A** Illustration depicting electrode and cannula placements. **B** Representative photomicrograph showing injector tip placement in the RE. Similar photomicrographs were taken to confirm placements for all animals. **C** Behavioral timeline. **D** Average freezing behavior (*n* = 11; 8 male and 3 female) across fear conditioning and extinction. **E** Freezing behavior showing that muscimol inactivation of the RE impairs extinction retrieval. **F** Average theta-range spectral coherence between the

mPFC and HPC after either muscimol or saline infusion in the RE. Muscimol inactivation of the RE impaired IL-HPC 6−9 Hz coherence (*p* = 0.017, two-way ANOVA with Bonferroni correction). mPFC medial prefrontal cortex, HPC hippocampus, RE nucleus reuniens of the thalamus. Line plots represent mean ± s.e.m.s; boxplots represent mean plus minima and maxima with lower and upper quantiles. *\*p* < 0.05. Source data are provided as a Source Data file.

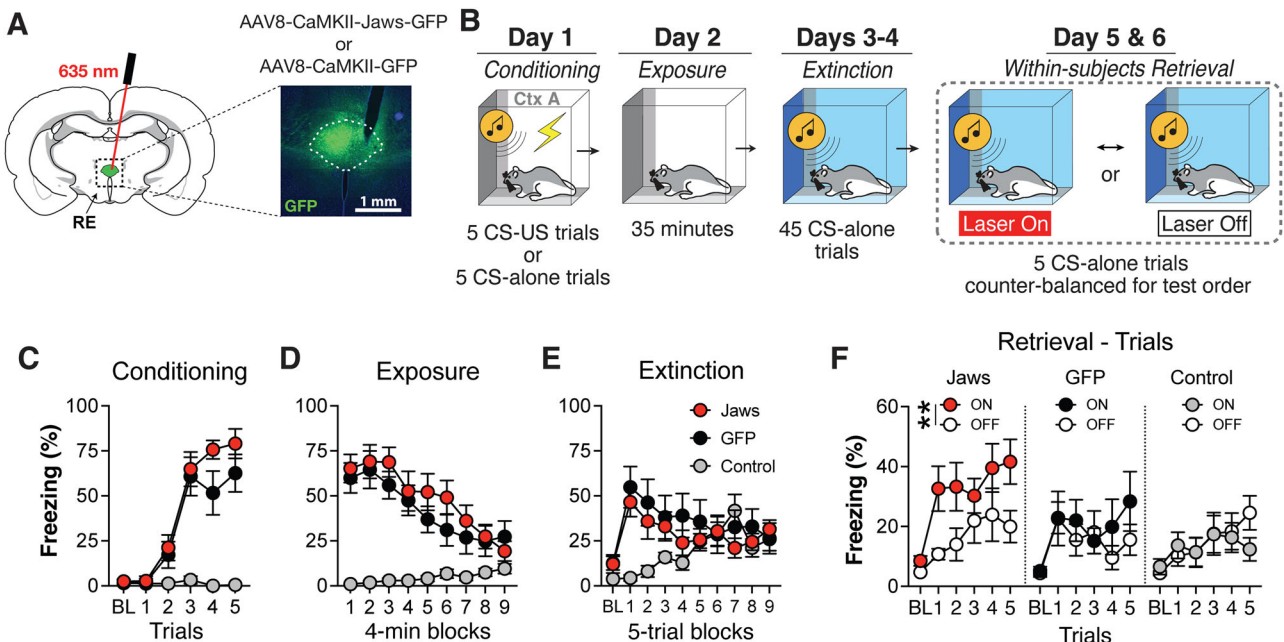

**Fig. 5 | Optogenetic inactivation of the RE selectively impairs extinction memory retrieval.** **A** Illustration depicting fiber placement and intracranial injections of either the active Jaws (AAV8-CaMKII-Jaws-GFP) or GFP control virus (AAV8-CaMKII-GFP) into the RE. **B** Experimental timeline. Average freezing behavior during fear conditioning (**C**), context exposure (**D**), and extinction (**E**). **F** Average freezing behavior during extinction retrieval showing that optogenetic inhibition of

the RE selectively increases freezing in conditioned animals expressing the active Jaws virus (*n* = 12; 5 male and 7 female; *p* = 0.0044, two-way ANOVA with Bonferroni correction), but not GFP (*n* = 10; 6 male and 4 female; *p* = 0.969) or non-conditioned control animals (*n* = 8; 5 male and 3 female; *p* > 0.999). RE nucleus reuniens of the thalamus, GFP green fluorescent protein. Line plots represent mean ± s.e.m.s. *\*\*p* < 0.01. Source data are provided as a Source Data file.

that 8-Hz sinusoidal stimulation can entrain RE LFPs and single units at theta frequency.

During behavioral testing, rats expressing the light-sensitive opsin (ChR2; *n* = 13; 9 male and 4 female) and control rats (mCherry; *n* = 13; 5 male and 8 female) acquired similar levels of conditioned freezing (Fig. 6M; main effect of Trials: $F_{5, 120} = 36.52.90$, *p* < 0.0001; Trials x Group interaction: $F_{5, 120} = 0.628$, *p* = 0.683) and extinguished freezing to the CS to similar degrees (Fig. 6M; main effect of Blocks: $F_{5, 216} = 10.5$,

*p* < 0.0001; Blocks x Group interaction: $F_{9, 216} = 0.521$, *p* = 0.859). If optogenetic stimulation of the RE blocks fear renewal, we would expect the ChR2 group to display lower freezing relative to mCherry controls during the blue-laser stimulation. Indeed, we found that ChR2-expressing rats displayed decreased freezing levels when tested with the laser on, whereas the mCherry group showed increased freezing when the laser was on (Fig. 6N; Laser x Group interaction: $F_{1, 20} = 9.70$, *p* = 0.0055; Trial × Laser × Group interaction: $F_{5, 100} = 3.99$,

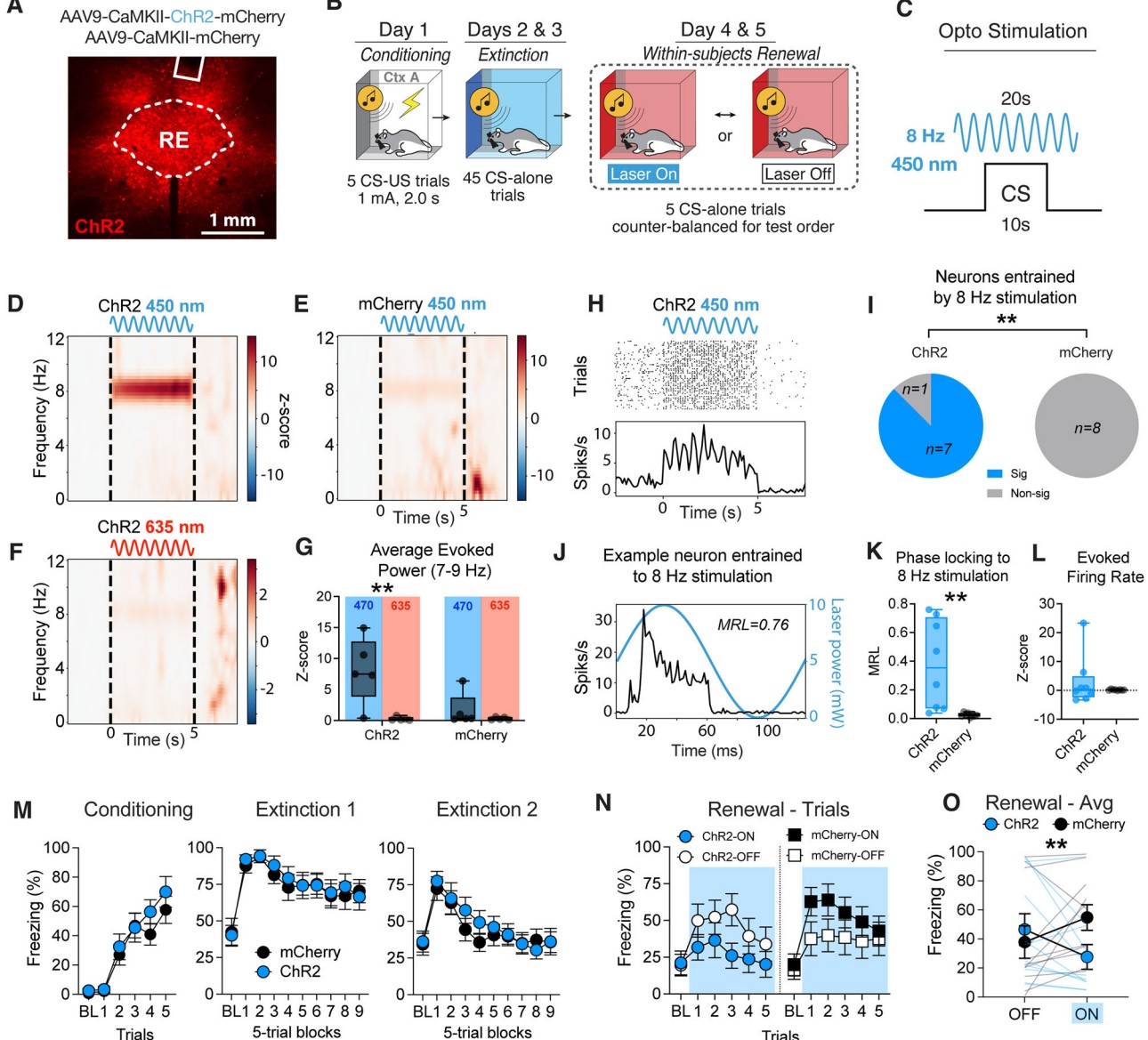

**Fig. 6 | Sinusoidal 8-Hz stimulation of the RE attenuates the renewal of extinguished fear. A** A representative image showing viral expression and optic fiber placement targeting the RE. Similar photomicrographs were taken to confirm placements for all animals. **B** Experimental timeline for fear renewal procedure. **C** Schematic of the optogenetic stimulation protocol by which RE neurons were stimulated with an 8-Hz sine wave pattern beginning five seconds before CS onset and stopping five seconds after CS termination. **D** Spectrograms showing that blue-laser stimulation using 8-Hz sine waves induces 8-Hz rhythms in the RE LFPs of ChR2-expressing rats, (**E**) but not control mCherry-expressing rats. **F** Red-laser stimulation was also insufficient to induce 8-Hz rhythms in ChR2 rats. **G** Average laser-evoked theta power was greatest in ChR2 under blue-laser stimulation, relative to red-laser ($n = 5$/group, $p = 0.0014$, two-way ANOVA with Bonferroni correction). **H** Raster plot of one example single unit phase-locked to blue-laser stimulation in a ChR2-expressing rat. **I** 8-Hz sinusoidal stimulation entrained 7/8 RE units in ChR2-expressing rats, compared to 0/8 units in mCherry-expressing rats

($n = 8$ units/group, $p = 0.0014$, Fisher's exact test). **J** Average firing rate in relation to the phase of laser stimulation of the same example neuron in H. **K** 8-Hz sinusoidal stimulation strongly entrained the firing of RE neurons ($n = 8$ units/group, $p = 0.0061$, Unpaired $t$-test), (**L**) despite not uniformly increasing their firing rates ($p = 0.374$, Unpaired t-test). **M** Average freezing data during conditioning and extinction sessions. **N** Freezing behavior during fear renewal trials shows that sinusoidal 8-Hz stimulation of RE attenuates fear renewal ($p = 0.006$; Three-way ANOVA, Laser x Group interaction) in rats expressing ChR2 ($n = 11$; $p = 0.0493$, Bonferroni correction) and not mCherry ($n = 11$; $p = 0.0835$, Bonferroni correction). **O** This is further confirmed by comparing the average freezing across the five CS trials ($p = 0.004$; Two-way ANOVA, Laser × Group interaction). RE nucleus reuniens of the thalamus, MRL mean resultant length. Line plots represent mean ± s.e.m.s; boxplots represent mean plus minima and maxima with lower and upper quantiles. *$p < 0.05$; **$p < 0.01$. Source data are provided as a Source Data file.

$p = 0.0024$). To further confirm these results, we compared the averages of the five CS trials of each group across laser conditions (Fig. 6O), which confirmed the Group x Laser interaction ($F_{1, 20} = 10.61$, $p = 0.004$). Post-hoc comparisons using the Bonferroni method revealed that 8-Hz stimulation significantly decreased freezing in ChR2-expressing rats ($p = 0.049$), whereas the increase displayed by mCherry rats just missed significance ($p = 0.084$). We speculate that

the moderate increase in freezing observed in mCherry rats is likely due to external disinhibition driven by the visually perceptible laser stimulus[46], effectively driving further renewal. This effect was attenuated by RE stimulation in ChR2-expressing rats. Importantly, optogenetic stimulation prior to CS presentations had no effect on freezing behavior (SFig. 6), and there were no test-order effects for any of the within-subject experiments reported in this paper (SFig. 7). We thus

conclude that 8-Hz sinusoidal stimulation of the RE attenuates the renewal of extinguished fear.

Based on findings showing that the RE displays 4-Hz rhythms during high fear states (Fig. 2), we additionally tested to see if 4-Hz sinusoidal stimulation of the RE was sufficient to induce fear relapse in the extinction context. We found that 4-Hz stimulation did not affect freezing in either the ChR2 or mCherry groups, demonstrating that 4-Hz stimulation is insufficient to drive fear renewal (SFig. 5).

## Discussion

Understanding the neural mechanisms that mediate fear suppression is critical to improving therapeutics for anxiety- and trauma-related disorders[2,14]. Previous work found that the RE and its prefrontal afferents are necessary for the retrieval of fear extinction memories[33,34,47], however, the mechanism by which the RE mediates this fear suppression remains unknown. Here, we show that fear extinction is characterized by enhanced mPFC-HPC theta synchrony and theta-range oscillations in the RE during fear expression and extinction. Inactivation of the RE-impaired mPFC-HPC network activity (c-Fos and theta synchrony) underlying extinction memory retrieval and theta-paced stimulation of the RE rescued context-dependent fear renewal. Collectively, this work shows that the RE is critical for PFC-HPC theta synchrony that accompanies extinction memory retrieval.

Substantial work has now demonstrated that limbic brain regions display dissociable 4-Hz and 8-Hz theta-range oscillations during freezing and non-freezing behavior, respectively[16,22,23,48–50]. The slower 4-Hz rhythms arise from respiratory patterns that act to entrain the olfactory bulb, and downstream mPFC and amygdala, to maintain high fear states[48,51–53]. In line with this, we found that increases in 4-Hz power in both the mPFC and RE strongly correlated with increases in freezing behavior throughout fear extinction and retrieval. Interestingly, despite the prominence of 4-Hz rhythms in the mPFC and RE, we found that mPFC-HPC synchrony during fear retrieval and extinction was specific to 8-Hz theta oscillations. This is in line with higher frequency "type-I" hippocampal theta commonly observed in the dorsal HPC during movement, whereas lower frequency "type-II" hippocampal theta is more commonly associated with the ventral HPC (vHPC) during immobility[54–56]. Indeed, the vHPC has been shown to play a more prominent role in processing motivationally relevant information[57–59] and selectively synchronizes with the mPFC and amygdala during negatively valenced behaviors[60–64]. Interestingly, it was recently shown that the vHPC synchronizes with and leads the mPFC during fear memory retrieval[65], the opposite directionality that might be predicted during extinction retrieval[18,33]. Moreover, direct projections from the vHPC provide feedforward inhibition of mPFC neurons to drive fear renewal[12], a relapse phenomenon not dependent on the RE[33]. Collectively, this provides support for the idea that bidirectional HPC→mPFC and mPFC→RE → HPC interactions act to guide context-appropriate recall of episodic memories[9,66,67], and that these interactions may be facilitated by neural synchrony in different frequency bands[41,68–70].

Fear relapse associated with RE inactivation may be due to the impaired ability of rats to successfully inhibit fear memories in the extinction context. We and others have previously shown that inactivating the RE prior to extinction retrieval causes fear relapse[33,34,47]. However, it was recently found that RE inactivation may in itself be anxiogenic[42], which might account for increases in freezing behavior after RE inactivation. However, we found that increased freezing after RE inactivation was only exhibited by animals that underwent conditioning and extinction; RE inactivation did not increase freezing in non-conditioned animals. Moreover, pharmacological inactivation of the RE does not increase freezing during baseline periods prior to CS presentations[33]. Although we did not observe differences in freezing during the baseline period, RE activity during the baseline period may be critical for extinction retrieval. Previous work has found that the RE is necessary for properly discriminating contexts[35], and reductions in

context discrimination might result in a relapse of fear[43]. Nonetheless, we found that optogenetic inactivation of RE during CS trials was still sufficient to drive fear relapse. Moreover, we found that 8-Hz sinusoidal stimulation of the RE was sufficient to drive extinction retrieval in a novel context, which normally drives fear renewal. Collectively, this work demonstrates that fear relapse induced by RE inactivation is specific to memory-retrieval deficits and suggests that the RE mediates successful extinction retrieval, not via recognition/encoding of the extinction context, but rather via a CS-related mechanism.

Bidirectional prefrontal-hippocampal interactions are critical for several mnemonic functions, including the encoding and recall of episodic memories[9]. It has been hypothesized that prefrontal top-down control of hippocampal activity via the RE is critical for the suppression of context-inappropriate memories[8]. After extinction, the reduction of conditional freezing behavior in the extinction context requires that animals suppress retrieval of fear memories—memories that normally show strong generalization to any context in which an aversive CS is encountered[4]. Previous work has shown that theta oscillations in the mPFC lead those in the HPC during extinction retrieval[18] and we have shown that the RE and its afferent projections from the mPFC are critical for extinction encoding and retrieval[33]. The present work reveals that RE coordinates oscillatory synchrony in the mPFC and HPC and that RE inhibition reduces mPFC-HPC coherence and attenuates extinction retrieval. Moreover, theta-paced stimulation of RE can restore extinction retrieval and attenuate fear relapse. These data suggest that mPFC-HPC interactions mediated by the RE are essential to the retrieval operations that permit the suppression of context-inappropriate fear memories in the extinction context. Given that the HPC has also been shown to encode contextual extinction memories[71,72], it is also conceivable that this circuit is involved in arbitrating the competition between fear and extinction memories for expression in behavior. Given that the RE exerts a predominantly inhibitory influence over the HPC[38] and RE inactivation leads to fear relapse, we speculate that the mPFC→RE → HPC projections function to suppress the retrieval of hippocampal fear memories, thereby promoting the suppression of fear in the extinction context.

In summary, the experiments reported here demonstrate that the retrieval of fear extinction memories requires mPFC-HPC interactions that are coordinated by the RE. This work expands our neurobiological understanding of the role of the prefrontal-hippocampal interactions in fear suppression and we suggest that the RE may be a potential therapeutic target for memory-based disorders, such as PTSD.

## Methods
### Animals
Male and female adult Long-Evans Blue Spruce rats, initially weighing between 200 - 240 g (~8 weeks old), were sourced from Envigo in Indianapolis, IN. Each rat was accommodated individually in transparent plastic enclosures within a temperature-regulated vivarium. The environment followed a steady 14:10 h light to dark pattern, with illumination beginning at 7:00 AM. Experiments were exclusively performed during the illuminated period. The rats had unrestricted access to food and water. For at least 5 days before any surgical or behavioral actions commenced, experimenters interacted with the rats, handling them for about 30 s each day. All the experimental methodologies adhered to the guidelines set by the US National Institutes of Health (NIH) for the Care and Use of Laboratory Animals and received approval from the Institutional Animals Care and Use Committee (IACUC) at Texas A&M University.

### Viruses
AAV9-CaMKII-hChR2(H134R)-mCherry, AAV9-CaMKII-mCherry, AAV8-CaMKII-Jaws-KGC-GFP-ER2, and AAV8-CaMKII-GFP viruses were purchased from Addgene (Watertown, MA). All viruses were diluted to a final titer of ~4–6 × 10$^{12}$ GC/mL with sterilized 1x DPBS.

## Surgery

Similar to our previous work[73], rats were anesthetized during surgical procedures using isoflurane, with a 5% concentration for induction and 1–2% for maintenance, and then positioned in a stereotaxic apparatus (Kopf Instruments). In a designated area, their scalp hair was trimmed, the skin was treated with povidine-iodine, and an eye ointment was applied. A minor incision was made on the scalp to reveal the skull's surface. The skull was then aligned such that the bregma and lambda points were on the same horizontal axis, and burr holes were drilled for the placement of intracranial tools including electrodes, cannulae, and optical fibers, as well as four small Jeweler's screws.

In the case of optogenetic studies, rats were given bilateral infusions of viruses (0.5 µl each) into the RE. These infusions were administered at a 0.1 µL/min pace, and the injector tips remained in the brain for ten minutes to facilitate diffusion. The designated coordinates for these viral infusions into the RE were AP: −2.10 mm, ML: −1.25 mm, DV: −7.09 mm, angled at 10° relative to the bregma skull surface. Right after this procedure, optical fibers with white ceramic ferrules were implanted into the RE, 0.3 mm above the site of the viral injection. These fibers were sourced from Thorlabs (Newton, NJ) and had a 200-µm core with a 10-mm length.

For recording electrophysiological data, a 16-channel microwire array from Innovative Neurophysiology (Durham, NC) was permanently implanted, targeting the mPFC and the dorsal CA1 area of the HPC always in the right hemisphere. The mPFC-targeted microelectrodes were constructed of 16 tungsten wires arranged in a 2×8 matrix, with one row having eight electrodes of 10.5-mm length and the other having a length of 5-mm. This design ensured simultaneous targeting of the infralimbic and prelimbic cortices. The HPC was targeted using a 4×4 array of 5-mm length. All wires were spaced 200-µm apart and each wire measured 50-µm in diameter. The ground and reference channels were linked, and a single silver grounding wire was coiled around a skull screw above the cerebellum, secured with conductive silver paint. The AP and ML coordinates were centered on the array and the array's leftmost wire, respectively. Dental adhesive was used to fix both optical fibers and microelectrode grids to the skull. In experiments that combined electrophysiological and optogenetic methods, 4-channel optrodes (Opto-MWA; Microprobes, Gaithersburg, MD) were employed. These optrodes were made up of four 10-mm stainless steel electrodes and a single 9.5 mm optic fiber, which was situated ~500 µm above the electrode ends. Optrodes were implanted three weeks post-viral infusions.

After surgery, a topical antibiotic (Triple Antibiotic Plus, G&W Laboratories) was applied to the incision area. Additionally, for pain relief post-surgery, rats were given a subcutaneous carprofen injection (5 mg/kg). A recovery period of at least a week was ensured before initiating any behavioral procedures.

## Behavioral procedures

Fear conditioning and extinction was carried out in two distinct rooms. Each conditioning chamber contained aluminum side panels, a Plexiglas top and rear, a hinged Plexiglas door, and a grid-based floor. This grid floor had 19 stainless steel rods connected to a shock delivery mechanism and a grid scrambler (Med Associates). Each compartment was equipped with a speaker for auditory cues, ventilation systems, and integral lighting. Fear retrieval tests took place in similar chambers that either had a red laser (Dragon Lasers, Changqun, China) or an electrophysiological system (Plexon, Dallas, TX).

For conditioning and optogenetic experiments, locomotor movements were monitored in real-time using a load-cell system under behavior compartment. This system transformed chamber movements into electrical readings via the Threshold Activity software from Med-Associates. In electrophysiological tests, this load-cell activity was directly captured by the OmniPlex software from Plexon, which allowed for simultaneous tracking of movement and electrophysiological data. In all setups, 'freezing' was marked as a stationary state that persisted for at least one second.

Three distinct environmental contexts, denoted as A, B, and C, were utilized during the conditioning and retrieval stages.: Context A had the fans, overhead, and house lights turned off. The chamber's cabinet door remained open throughout the session. Before each squad, the chamber was cleansed with 1.0% ammonium hydroxide and rats were transferred to this context in black plastic containers. In Context B, the chamber's house light and fans were turned on, with the environment dimly illuminated by overhead red fluorescent lights. The cabinet door was kept closed for the entirety of the session. A black Plexiglas layer was placed over the grid floor. The chamber was cleaned using a 3.0% acetic acid solution prior to sessions. Rats were moved to this context in white plastic containers with fresh bedding. For Context C, both the house and overhead lights were turned on, the fan was off, and the cabinet door stayed shut. The black and white striped paper was placed on the chamber walls, with a clear plastic floor covering the grid. The chamber was cleaned with 70% ethanol before any session, and rats were moved in white plastic containers with fresh bedding.

In all behavioral tests, rats underwent a standard auditory fear conditioning process in context A, where a non-threatening auditory tone (conditioned stimulus; CS) was paired with a distressing footshock (unconditioned stimulus; US). This included a 3-min stimulus-free baseline period, followed by five CS (10 s, 80 dB, 8 kHz) and US (1.0 mA, 2 s) pairings with 60-s intervals between trials (ITIs) and an extra 60-s post-shock phase. Unless specified otherwise, all rats experienced fear extinction in context B, 24 h later. This consisted of a 3-min baseline and then 45 solo CS presentations without the US (10 s, 80 dB, with 30 s ITIs). Some rats, showing resistance to fear extinction, underwent multiple extinction sessions until their fear responses were acceptably low. In experiments involving optogenetics and drugs, rats underwent a retrieval session that exposed them to Laser/Drug conditions in a counterbalanced sequence.

Electrophysiological validation of optogenetic stimulation took place in an opaque 5-gallon bucket. Prior to the fear conditioning and extinction phases, rats underwent an optogenetic stimulation regimen involving either 4- or 8-Hz sine waves with blue light (450 nm) for 5 s and 15-s ITI phases. The following day, 8 or 4 Hz stimulations were administered in a balanced sequence. After behavioral testing procedures, they were again tested with a red laser (635 nm) using the same within-subject strategy to eliminate potential light-induced anomalies. Both blue and red lasers were adjusted to output 10 mW at the fiber tips before any testing commenced. More information is available in the "Optogenetics" methods section.

## Drug infusions

For drug microinfusions, rats were moved into a room adjacent to the vivarium and placed into 5-gallon white buckets. Dummy cannula internals were removed and a stainless-steel injector (33-gauge, 9 mm; Plastics One) connected to polyethylene tubing was inserted into the guide cannula. Polyethylene tubing was connected to 10-µl Hamilton syringes that were mounted in an infusion pump (Kd Scientific). Muscimol was diluted to a concentration of 0.1 µg/µl in sterile saline. Infusions were made at a rate of 0.3 µl/min for 1 min and the injectors were left in place for 2 min post-infusion to allow for adequate diffusion. Each infusion was verified by the movement of an air bubble that separated the drug or sterile saline from distilled water within the polyethylene tubing. Clean dummy internals were inserted into each guide cannula after infusions. All infusions were made ~5 min prior to behavioral testing. All animals were acclimated to these procedures by performing dummy changes twice before beginning behavioral procedures.

## Optogenetics

For experiments using the red-light activated opsin, Jaws, a red laser (Dragon Lasers) with a wavelength of 635 nm was used. For experiments using the blue-light activated opsin, ChR2, a blue laser (Dragon Lasers) with a wavelength of 450 nm was used. All the optic fibers used were hand crafted and only those displaying an efficiency exceeding 80% post-polishing were selected. For every experiment, the laser power was calibrated to provide 10 mW at the end of the patch cord termination, ensuring an approximate power of 8–9 mW at the optical fiber tips. To facilitate optostimulation without restricting the rats' natural movement or exploratory actions, a fiber-optic rotary joint from Doric Lenses and a bundled patch cord from Thorlabs were used. In experiments designated as "constant" inhibition, Med Associates software interfacing with a transistor-transistor logic (TTL) adapter controlled the laser. This setup ensured light exposure began 10 s prior to the first CS initiation and continued until the session's conclusion. 8-Hz sinusoidal stimulation pattern was generated using a DG812 Waveform Generator from RIGOL Technologies, Inc., which was driven by TTL triggers from Med Associates software.

## Electrophysiology

Electrophysiological recordings and freezing behavior were captured using the OmniPlex system from Plexon (Dallas, TX.) Local field potentials (LFP) were recorded from a single wire referenced to a skull screw, with other channels referenced to the LFP wire, enhancing single-unit recordings. Data was captured at 40k Hz, amplified 3000x, and stored for future analysis. After high-pass filtering at 600 Hz, candidate waveforms underwent manual sorting using the first two principal components in principal component analysis (PCA) space via the Offline Sorter (Plexon). Only distinctly separated units in PCA space were analyzed, and duplicate units on neighboring electrodes were excluded from downstream analyses.

Local field potentials were down-sampled to 1000 Hz prior to analysis. Analyses were conducted using a combination of custom-written MATLAB and Python scripts. All data were detrended and notch filtered at 60 Hz (58–62 Hz window) to remove power line noise. The raw data for all trials were plotted and any trials with noticeable motion artifacts were manually removed from all analyses. For power analyses, power spectral density (PSD) estimates were calculated using Welch's method (*pwelch* MATLAB function). PSDs were then averaged across trials to obtain one PSD estimate per subject for each time point of interest. PSDs were then normalized to the total power of 2–12 Hz (i.e. relative power) to account for differences in total signal strength between animals. Average power in frequency bands of interest (3–6 Hz and 6–9 Hz) were then averaged to obtain a single frequency-band power estimate for each subject. For coherence analyses, the magnitude squared coherence was also calculated using Welch's method (*mscohere* Matlab function) and the average peak coherence was determined by taking the maximum value in the frequency band of interest. Coherograms were created using the multi-taper method from the Chronux toolbox using a 3 s moving window with 100 ms overlap. Evoked spectral power was determined by computing the baseline normalized Z-score for the average power, using the 2 s pre-laser period as the baseline.

Single-unit analyses were carried out using custom-written Python scripts. Peri-event time histograms (PETHs) were generated using the *compute_perievent* function from the Pynapple package[74] using a bin size of either 100 ms or 1.25 ms. Raleigh's test of uniformity was used to determine if single units were significantly entrained to the phase of sine wave stimulation. Evoked firing was determined by computing the baseline normalized Z-score for the average firing rate, using the 2 s pre-laser period as the baseline.

## Histology

After the end of behavioral experimentation, rats were sacrificed to confirm electrode, cannula, viral, and/or fiber optic placement. To help visualize electrode tip placements, a small current (0.1 mA) was passed through the four of the sixteen microelectrodes for 10 s to create a small lesion immediately before transcardial perfusions. Animals were overdosed with sodium pentobarbital (Fatal Plus, 100 mg/ml, 0.7 ml), transcardially perfused with ice-cold saline and fixed with 10% physiological formalin.

Perfused brains were placed in physiological formalin for 14–24 h before being moved to a 30% sucrose solution for a minimum of three days. After three days, or until brains had sunk, all brains were frozen and sectioned at −20° on a cryostat. To verify cannula and electrode placements, brains were sectioned at 40 μm, mounted onto gelatin subbed slides, thionin stained (0.25%), cover-slipped with Permount (Fisher Scientific), and then imaged on a wide-field stereoscope. To confirm viral expression and optic fiber localization, brains were sectioned at 30 μm, mounted onto gelatin subbed slides, cover-slipped with DAPI-infused Fluoromount (Fisher Scientific), and then imaged on a 10x fluorescent microscope.

## Fos immunohistochemistry

All rats were overdosed with sodium pentobarbital and transcardially perfused. Brains were dissected and stored in 10% formalin for up to 14 h and then transferred to 30% sucrose at 4 °C for at least 72 h. After all brains were sectioned on a cryostat (−20°) at 30 μm and brain sections. For Fos immunohistochemistry, brain sections were washed 3x in TBST, incubated in 0.3% H20h for 15 min, washed 3x in TBS, and then incubated in rabbit anti-c-fos primary antibody (1:1000; Millipore) overnight. The next day the sections were washed 3x in TBS followed by a 1-h incubation in a biotinylated goat anti-rabbit secondary antibody (1:1000, Jackson Immunoresearch), amplified with the avidin biotin complex at 1:1000 (ABC; Vector labs), and visualized with 3,3'-diaminobenzidine (DAB) + nickel ammonium sulfate. Stained sections were mounted in subbed slides, coverslipped with Permount and stored at room temperature until they were imaged.

To quantify c-Fos expression, one 10X image (895 μm × 670 μm; 0.596 mm$^2$) was taken of each hemisphere at different AP levels of each the PL and IL (+2.7 mm and +2.3 mm from bregma), HPC (−5.5 mm and −6.0 mm), and MGN (−5.5 mm and −6.0 mm). The total number of Fos-expression neurons in each image was manually counted, averaged across all images for each image, and divided by the surface area to the average number of Fos cells/mm$^2$.

## Statistics

All data were analyzed using custom-written Matlab and Python scripts. Statistical analyses of the data were performed using GraphPad Prism (version 9.0; GraphPad Software). Male and female rats were collapsed as there were no sex differences in any of the analyses. Data were analyzed with repeated-measures analysis of variance, and significant interactions were followed by Tukey's multiple comparisons test unless otherwise noted. Group sizes were determined based on our prior work.

## Reporting summary

Further information on research design is available in the Nature Portfolio Reporting Summary linked to this article.

# Data availability

The data for each figure shown in this manuscript are included in the Source Data files. Raw data from these experiments are available from the corresponding author upon request. Source data are provided with this paper.

# Code availability

All scripts used to analyze the data in this manuscript were written and executed in either MATLAB 2022a or Python 3.9.12 and are available from the corresponding author upon request.

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

## Acknowledgements

We thank Dr. Jianfeng Liu for his assistance with the optogenetic experiments. We also thank Drs. Justin Moscarello, Rachel Smith, and Ursula Winzer-Serhan for helpful discussions around the direction and interpretation of this work. This work was supported by the National Institutes of Health (R01MH065961 and R01MH117852 to S.M.).

## Author contributions

M.S.T., T.T. and S.M. designed the electrophysiological and optogenetic experiments; K.R.R., J.J. and S.M. designed the c-Fos experiment. K.R. and J.J. performed c-Fos immunization and counting. M.T., T.T. and S.E.P. performed optogenetic experiments; M.T. and T.T. analyzed the data. M.T. performed all other experiments and analyzed the electrophysiological data. M.T. and S.M. wrote the manuscript.

## Competing interests

The authors declare no competing interests.
