## [Peer Review File · Nature Communications]

Thalamic nucleus reuniens coordinates prefrontal-hippocampal synchrony to suppress extinguished fearREVIEWER COMMENTS

Reviewer #1 (Remarks to the Author):

This manuscript reports a series of findings supporting the hypothesis that the nucleus reuniens is necessary for mPFC-HPC synchronization during the retrieval of extinction memories. The research question is important because therapies for pathological fear can only be effective if fear memories are weakened, likely through fear extinction. Re could thus be a potential target for therapeutic interventions for pathological fear.

The studies are methodologically sound with adequate detail provided in most cases.

One issue that should be addressed is whether the differences in delta and theta power & coherence can be explained by differences in the rats' movement between time points.

Another issue is that throughout the results, they use the word correlated, but in the discussion, they claim that "RE mediates fear memory suppression by synchronizing mPFC-HPC theta oscillations" (lines 340-341). I'm not convinced that the correlation results lead to that strong of a conclusion.

Here are some suggestions for improving the clarity of the manuscript.

Line 69: typo

Line 70: use Re instead of reuniens (already defined)

Line 73: what is the hypothesis about the direction of information flow between mPFC and Re?

Implying that this pathway is suppressing the fear memory? How would this work? Bring in the idea that Re projections to hippocampus have an inhibitory influence?

Line 85: The controls do not "demonstrate". The way that the sentence is written is incorrect. The control animals don't demonstrate...Instead, the experimental results from the control animals demonstrate...Revise grammar.

Line 97: When reporting N's throughout the paper, please report how many males and females were used.

Figure 1 caption: Does B=baseline? "B" should be defined.

Figure 1 and text describing Fig. 1 results: Add details about how the PSDs were normalized.

Figure 1N: Add indication of the time axis.

Figure I&L: bar colors indicate same timepoints as panel F?

Figure 1P: Peak coherence is not defined or explained in the text or method section.

Figure 1O: is this an example from one rat/session?

Figure 1E: why did freezing drop on the last conditioning day?

Line 142 and multiple other times throughout the manuscript, the authors start a sentence with "This..." without defining what "this" is.

Line 157: similar to what?

Line 159: change "off" to "to"

Figure 2 caption: explain "context A vs B".

Line 184: confusing wording. Maybe delete the word "different"

Line 189: "(refs)" I assume that you forgot to add specific references here.

Line 203: Why was the MGN chosen as a control region?

Figure 3 caption: BL=baseline? If so, be consistent with Fig. 1

define what "First" and "Final" mean.

Also, add a legend so it is clear that red=Mus and black = Sal.

Figure 3caption (B) is "impair" the correct word?

Line 223: move this description of muscimol to the section above, where it is first mentioned.

Figure 4B caption: how does this photo show spread?

Line 241: is impair the appropriate word?

Line 260: Refer to Fig. 5B

Figure 5 panel labels are incorrect here and in the text.

Line 280: Specify the control group to which you are referring.

Figure 6 caption: Describe how G and H are different.

How were the values calculated?

Line 327: Provide more details about how the values were calculated.

Line 398: I suggest working this idea into the Introduction section.

Caveat is that this is currently a preprint and has not been replicated. It also goes against Van der Weel's findings.

Line 524: what are TTL riggers?

Reviewer #2 (Remarks to the Author):

In this manuscript, Totty and colleagues studied the roles of the prefrontal cortex-thalamic nucleus reuniens-hippocampus circuits for extinction of fear memory. This study was built on the group's previous work. In this study, the authors conducted in vivo extracellular recording, pharmacological and optogenetic silencing, and optogenetic stimulation to firmly establish the critical role of nucleus reuniens in connecting the prefrontal cortex and the hippocampus in fear extinction. The topic should have broad audience in the field of neuroscience. The experiments were well designed and executed. The results were clearly presented in the figures. The overall quality of this manuscript is comparable to the very good ones published in Nature Communications on similar topics. I only have a few minor issues:

1) EEG recording was done with the DORSAL hippocampus (Fig1, 4), while the C-fos quantification was done with the VENTRAL hippocampus (Fig.3). In the Discussion, the authors also emphasized the synchrony between the prefrontal cortex and the ventral hippocampus. Considering the functional differences between the dorsal and ventral hippocampus, a rationale is needed for this experimental design. The authors may also want to discuss the implications of this in the interpretation of the data.

2) Interpretation of the correlation of oscillation patterns with freeze levels. It is a fascinating finding that the extinction of fear (decreases in freeze levels) to tone is associated with a shift of 3-6 Hz to a 6-9 Hz oscillation in the prefrontal cortex. The 6-9 Hz oscillation has the highest power when the animals were exposed to a novel context (Expo), while the 3-6 Hz has highest power

when the animals reacted most to the tone. Could the shift in theta frequency at the prefrontal regions reflect the processing of contextual vs. auditory information? In other words, could the correlation of freeze levels to theta powers just reflect that animals preferentially attended to the context instead of the tones? In fact, in Fig. 1J and 1M, the correlation only existed when the data from the three different stages were plotted together.

3) Validation of the techniques. The sine wave stimulation of nucleus reuniens is interesting, but no evidence shows that this optogenetic stimulation indeed mimicked the theta oscillation in this region. Similarly, the authors did not show the c-fos data in the nucleus reuniens to demonstrate the effectiveness of muscimol (fig.3)

4) The authors wrote in Discussion that "Given that the RE exerts a predominantly inhibitory influence over the HPC, we speculate that the RE acts to suppress the retrieval of hippocampal fear engrams, thereby facilitating the retrieval of hippocampal extinction engrams." What exactly does "engram" mean here (for context or for tone)? How can the suppression/facilitation be selectively achieved?

5) Minor: In Fig1.E, the authors can consider to color code the data points as that in the following panels (1F-1M), so that readers can easily connect the data presented in different figure panels. There are a few typos, and references planned but not added.

Reviewer #3 (Remarks to the Author):

The manuscript submitted by Totty and colleagues describes a series of experiments aimed at investigating the role of the nucleus reuniens in mediating: 1) synchronized activity between the prefrontal cortex and hippocampus, and 2) extinction-induced suppression of fear behavior. Previous studies have already established a role for the nucleus reuniens in both prefrontal-hippocampal synchrony and in fear suppression, but no previous study has directly connected these 2 functions of the nucleus reuniens. The study by Totty and colleagues for the first time directly connects these 2 functions by manipulating the nucleus reuniens in rats subjected to tone fear conditioning and extinction while recording both prefrontal-hippocampal synchrony and fear behavior. They found that within the same behavioral paradigm inactivating the nucleus reuniens causes a decrease in prefrontal-hippocampal synchrony and a decrease in extinction-induced fear suppression, while stimulation of the nucleus reuniens at the same frequency as the observed prefrontal-hippocampal synchrony promotes fear suppression. Combined, the results reported by Totty and colleagues provide an important contribution to the mechanistic understanding of extinction-induced fear suppression and to possible translational approaches for treating disorders associated with excessive fear.

The experiments, results, and conclusions are very clearly described. I do have some questions and suggestions.

1) The data shown in figures 4-6 were obtained using a counter-balanced design, with the data collapsed in the figures. Does this mean that no test-order effects were observed? It would be helpful to add supplemental figures where the data are not collapsed, so that readers can determine if there could be any interesting trends towards test-order effects.

2) In figure 6 the effects of 8Hz stimulation of the nucleus reuniens are reported. In panel 6H the effects are compared between Chr2- and mCherry-expressing rats. According to the legend this panel shows that "sinusoidal 8 Hz stimulation of RE reduces fear renewal in rats expressing Chr2 (n = 6) and not mCherry (n = 6)". Visual inspection of the panel suggests that this description might not fully capture all the effects, as the absolute effect of light ON in the mCherry group seems larger than in the Chr2 group (though in opposite direction). Is there an effect in the mCherry group or not? In general, it would be helpful if the statistical results for all the data shown throughout the manuscript, whether significant or not, were added in a supplemental table.

This would help readers understand how the conclusions presented throughout the manuscript were drawn.

3) Also related to figure 6H, if there is an effect of light ON in the mCherry group, then what is the explanation of a light effect in rats that do not express an opsin? The answer seems relevant to interpreting the effect in the ChR2 group, as the light effect in the ChR2 group seems relatively minor and appears to somehow negate the light effect in the mCherry group.

4) In figure 1N-P, coherence in the 6-9Hz range is shown. It would be helpful to add in a supplement the 3-6Hz range coherence data.

5) Similarly, in figure 4F the coherence is only shown in the 6-9Hz range, and it would be helpful to add the 3-6Hz range data in a supplement.

6) The optogenetic manipulations reported in figures 5-6 were initiated before the CS onset instead of at the CS onset, even though figure 2 shows that theta-range rhythms in the nucleus reuniens start at CS onset. It would be helpful if a rationale for this difference in timing is included, and also to include the freezing data from the time-interval preceding the onset of the first CS while the light was already turned on.

We would like to thank all of the reviewers for their constructive feedback and overall positive comments regarding our manuscript! Below you will find the **original reviewer comments in black**, **our responses in blue**, and **newly added text and figures from the manuscript in green**.

REVIEWER COMMENTS

Reviewer #1 (Remarks to the Author):

This manuscript reports a series of findings supporting the hypothesis that the nucleus reuniens is necessary for mPFC-HPC synchronization during the retrieval of extinction memories. The research question is important because therapies for pathological fear can only be effective if fear memories are weakened, likely through fear extinction. Re could thus be a potential target for therapeutic interventions for pathological fear.

The studies are methodologically sound with adequate detail provided in most cases.

One issue that should be addressed is whether the differences in delta and theta power & coherence can be explained by differences in the rats' movement between time points.

Previous work has indeed found that 4 vs 8 Hz power in limbic regions (PFC, HPC, and amygdala) are tightly correlated to behavioral states. Namely, 8 Hz theta power is strongly modulated by locomotion (PMID: 26455912, PMID: 33567253) whereas 4 Hz power is intrinsically tied to fear-related freezing behavior (PMID: 34790100; PMID: 33972521). In the current series of experiments, we replicate these findings as 4 and 8 Hz power are positively and negatively correlated with freezing behavior (Fig. 1J & M).

However, coherence between the PFC and HPC (which reflects PFC-HPC interactions) is more-so modulated by cognitive demand, such as memory encoding and retrieval (PMID: 28655882). Our current data nicely demonstrates this as PFC-HPC coherence is significantly different during extinction memory retrieval compared to simple context exposure; two timepoints of similarly lower levels of freezing (Fig. 1P), but different cognitive demands (Fig. 1F). To further demonstrate this point, we have added a new supplemental figure (**SFig 1**) showing that, unlike power, neither 6-9 Hz nor 3-6 Hz coherence is correlated to freezing behavior across experimental timepoints.

In summary, you are correct that the power of ~4 and 8 Hz oscillations may be explained by differences in freezing behavior. However, ~8 Hz coherence (which we argue reflects extinction memory retrieval) cannot be explained by differences in the rats movement between timepoints.

Supplemental Figure 1: PFC-HPC coherence cannot be explained by differences in freezing behavior. (A) Linear regression analysis showing that peak coherence in both the 6-9 Hz (A; non-significant slope: $F_{1, 16} =$

0.045, $p = .152$) and 3-6 Hz (B; non-significant slope: $F_{1, 16} = 0.079$, $p = .082$) frequency range has a weak relationship with the percentage of time animals exhibited freezing behavior across experimental timepoints.

Another issue is that throughout the results, they use the word correlated, but in the discussion, they claim that "RE mediates fear memory suppression by synchronizing mPFC-HPC theta oscillations" (lines 340-341). I'm not convinced that the correlation results lead to that strong of a conclusion.

We have tempered the language in lines 340-341 to more accurately reflect the main findings of the manuscripts. See below:

“Collectively, this work shows that the RE is critical for PFC-HPC theta synchrony that accompanies extinction memory retrieval.”

Here are some suggestions for improving the clarity of the manuscript.

Line 73: what is the hypothesis about the direction of information flow between mPFC and Re? Implying that this pathway is suppressing the fear memory? How would this work? Bring in the idea that Re projections to hippocampus have an inhibitory influence?

We have reinforced language in the introduction and discussion that inhibitory influences of RE on the hippocampus may be suppressing the retrieval of fear memories in the extinction context (in introduction:

“and inhibitory projections from the RE to HPC have been postulated to regulate memory retrieval”.

Line 97: When reporting N's throughout the paper, please report how many males and females were used.

The number of male and female rats used for each experiment is now stated where N's are reported.

Figure I&L: bar colors indicate same timepoints as panel F?

Yes, we have added legends to make this more clear.

Figure 1O: is this an example from one rat/session?

Yes, we have made this more clear in the figure and legend.

Figure 1E: why did freezing drop on the last conditioning day?

The X-axis for Conditioning in Fig 1. E is Trials, not Days. Although freezing does go up on average across trials, it is not uncommon to see a later trial that has slightly lower freezing during fear conditioning. This is often due to the rats exhibiting escape behavior rather than freezing.

Line 203: Why was the MGN chosen as a control region?

The MGN (auditory thalamus) is a site critical to auditory fear conditioning that has no direct conditions to the RE. Adding the MGN to this experiment controls for any potential changes in Fos not directly due to RE inactivation (i.e., changes in behavioral state or auditory processing). Because there were no changes in MGN Fos expression, we can conclude that changes in Fos expression in the PFC and HPC were directly due to RE inactivation. We have updated the manuscript to make this more clear.

As a control for non-specific effects we additionally quantified Fos expression in the medial geniculate nucleus (MGN), a thalamic region implicated in auditory fear conditioning with no direct connections to the RE. We found that MGN Fos was not affected by RE inactivation ($t_{27} = 0.71, p = .485$). This suggests that RE inactivation results in impaired neuronal activity in both the PFC and HPC.

Line 327: Provide more details about how the values were calculated.

Please see the details that we added to the Results section below.

“... we found that ChR2-expressing rats displayed decreased freezing levels when tested with the laser on, whereas the mCherry group actually showed increased freezing when the laser was on (Fig. 6N; Laser x Group interaction: $F_{1, 20} = 9.70, p = .0055$; Trial x Laser x Group interaction: $F_{5, 100} = 3.99, p = .0024$). To further confirm these results, we compared the averages of the five CS trials of each group across laser conditions (Fig. 6O), which confirmed the Group x Laser interaction ($F_{1, 20} = 10.61, p = .004$).”

Line 398: I suggest working this idea into the Introduction section.

This is a great recommendation and we have now suggested this possibility in the introduction.

“Additionally, inactivating the RE has been reported to impair mPFC-HPC synchrony and inhibitory projections from the RE to HPC are positioned to mediate retrieval suppression processes implicated in extinction.”

Line 69: typo

Line 70: use Re instead of reuniens (already defined)

Line 85: The controls do not "demonstrate". The way that the sentence is written is incorrect. The control animals don't demonstrate...Instead, the experimental results from the control animals demonstrate...Revise grammar.

Figure 1 caption: Does B=baseline? "B" should be defined.

Figure 1 and text describing Fig. 1 results: Add details about how the PSDs were normalized.

Figure 1N: Add indication of the time axis.

Figure 1P: Peak coherence is not defined or explained in the text or method section.

Line 142 and multiple other times throughout the manuscript, the authors start a sentence with "This..." without defining what "this" is.

Line 157: similar to what?

Line 159: change "off" to "to"

Figure 2 caption: explain "context A vs B".

Line 184: confusing wording. Maybe delete the word "different"

Line 189: "(refs)" I assume that you forgot to add specific references here.

Figure 3 caption: BL=baseline? If so, be consistent with Fig. 1

define what "First" and "Final" mean.

Also, add a legend so it is clear that red=Mus and black = Sal.

Figure 3caption (B) is "impair" the correct word?

Line 223: move this description of muscimol to the section above, where it is first mentioned.

Figure 4B caption: how does this photo show spread?

Line 241: is impair the appropriate word?

Line 260: Refer to Fig. 5B

Figure 5 panel labels are incorrect here and in the text.

Line 280: Specify the control group to which you are referring.

Figure 6 caption: Describe how G and H are different.

How were the values calculated?

Line 524: what are TTL riggers?

We have made changes according to all of the above suggestions. Thank you for your attention to detail.

Reviewer #2 (Remarks to the Author):

In this manuscript, Totty and colleagues studied the roles of the prefrontal cortex-thalamic nucleus reuniens-hippocampus circuits for extinction of fear memory. This study was built on the group's previous work. In this study, the authors conducted in vivo extracellular recording, pharmacological and optogenetic silencing, and optogenetic stimulation to firmly establish the critical role of nucleus reuniens in connecting the prefrontal cortex and the hippocampus in fear extinction. The topic should have broad audience in the field of neuroscience. The experiments were well designed and executed. The results were clearly presented in the figures. The overall quality of this manuscript is comparable to the very good ones published in Nature Communications on similar topics. I only have a few minor issues:

1) EEG recording was done with the DORSAL hippocampus (Fig1, 4), while the C-fos quantification was done with the VENTRAL hippocampus (Fig.3). In the Discussion, the authors also emphasized the synchrony between the prefrontal cortex and the ventral hippocampus. Considering the functional differences between the dorsal and ventral hippocampus, a rationale is needed for this experimental design. The authors may also want to discuss the implications of this in the interpretation of the data.

The cFos experiment was originally conducted as part of our previous work focusing on ventral HPC->mPFC connectivity in fear renewal. However, this work shifted to the role of the RE in coordinating mPFC-HPC communication. When the recording study was implemented, we wanted to build on the observations of mPFC-dorsal HPC synchrony already established by Lesting et al (PMID: 24204927) in extinction retrieval.

Considering that the RE projects to the stratum lacunosum moleculare of the CA1 region throughout the septo-temporal axis of the hippocampus (PMID: 24025745), we speculate that we would similarly observe decreased cFos expression in the dorsal HPC due to RE inactivation. We hypothesize that the RE acts to facilitate extinction retrieval by inhibiting fear-encoding neurons throughout the septotemporal axis of the hippocampus.

2) Interpretation of the correlation of oscillation patterns with freeze levels. It is a fascinating finding that the extinction of fear (decreases in freeze levels) to tone is associated with a shift of 3-6 Hz to a 6-9 Hz oscillation in the prefrontal cortex. The 6-9 Hz oscillation has the highest power when the animals were exposed to a novel context (Expo), while the 3-6 Hz has highest power when the animals reacted most to the tone. Could the shift in theta frequency at the prefrontal regions reflect the processing of contextual vs. auditory information? In other words, could the correlation of freeze levels to theta powers just reflect that animals preferentially attended to the context instead of the tones? In fact, in Fig. 1J and 1M, the correlation only existed when the data from the three different stages were plotted together.

Please see the response to Review #1. The proposal that 3-6 Hz vs 6-9 Hz rhythms reflect differential processing of cued vs contextual processing is an interesting idea. However, this is unlikely because animals were presented with the tones during both extinction and extinction retrieval, which resulted in divergent ~4 and 8 Hz power, respectively (Fig. 1I & 1L). As we detailed above in a response to Reviewer #1, it is now well-documented that ~4 Hz power in the PFC is directly linked to fear-related freezing behavior via breathing cycles and the olfactory bulbs. These 4 Hz rhythms act to entrain the PFC (as well as other limbic regions) and maintain freezing bouts (PMID: 34790100; PMID: 33972521). Conversely, ~8 Hz theta power in the HPC and PFC is strongly linked to locomotion (PMID: 26455912, PMID: 33567253). However, this is not true for PFC-HPC coherence at 8 Hz which we argue reflects extinction memory retrieval.

3) Validation of the techniques. The sine wave stimulation of nucleus reuniens is interesting, but no evidence shows that this optogenetic stimulation indeed mimicked the theta oscillation in this region. Similarly, the authors did not show the c-fos data in the nucleus reuniens to demonstrate the effectiveness of muscimol (fig.3)

Although we did not quantify c-Fos expression, we do show robust behavioral effects in Fig. 3 which speak to the effectiveness of RE inactivation via muscimol. These data replicate our previous findings that muscimol inactivation of the RE results in fear relapse. Moreover, we have an independent replication in this manuscript (Fig. 4) where we achieve the same behavioral effect accompanied by altered electrophysiological signals.

We have now included an additional experiment demonstrating that 8 Hz stimulation of the RE is capable of both driving 8 Hz oscillations and entraining single-units located in the RE (Fig 5). This is accompanied by control experiments showing that these electrophysiological effects are not the result of light-induced artifacts but are indeed specific to animals expressing the excitatory opsin (ChR2) coupled to blue-laser stimulation. Thank you for this suggestion, we feel that this final experiment has significantly strengthened and solidified the current findings. The primary findings of these validation experiments are now shown in Figure 6 (shown below). Additional data can be found in Supplemental Figures 2 and 3.

Figure 6. Sinusoidal 8-Hz stimulation of the RE attenuates the renewal of extinguished fear. (A) A representative image showing viral expression and optic fiber placement targeting the RE. (B) Experimental timeline for fear renewal procedure. (C) Schematic of the optogenetic stimulation protocol by which RE neurons were stimulated with an 8 Hz sine wave pattern beginning five seconds before CS onset and stopping five seconds after CS termination. (D) Spectrograms showing that blue-laser stimulation using 8 Hz sine waves induces 8 Hz rhythms in the RE LFPs of ChR2-expression rats, (E) but not control mCherry-expressing rats. (F) Red-laser stimulation was also insufficient to induce 8 Hz rhythms in ChR2 rats. (G) Average laser-evoked theta power across conditions. (H) raster plot of one example single unit phase-locked to blue-laser stimulation in a ChR2-expressing rat. (I) 8 Hz sinusoidal stimulation entrained 7/8 RE units in ChR2-expression rats, compared to 0/8 units in mCherry-expressing rats. (J) Average firing rate in relation to the phase of laser stimulation of the same example neuron in H. (K) 8 Hz sinusoidal stimulation did not significantly change the average firing rate of RE neurons, (L) despite strongly modulating their firing patterns. (M) Average freezing data during conditioning and extinction sessions. (N) Freezing behavior during fear renewal trials shows that sinusoidal 8 Hz stimulation of RE attenuates fear renewal in rats expressing ChR2 ($n = 6$) and not mCherry ($n = 6$). (O) This is further confirmed by comparing the average freezing across the five CS trials. All data are means \pm s.e.m.s; * $p < 0.05$; ** $p < 0.01$.

4) The authors wrote in Discussion that “Given that the RE exerts a predominantly inhibitory influence over the HPC, we speculate that the RE acts to suppress the retrieval of hippocampal fear engrams, thereby facilitating the retrieval of hippocampal extinction engrams.” What exactly does “engram” mean here (for context or for tone)? How can the suppression/facilitation be selectively achieved?

We have removed the reference to “engram” here insofar as we are ultimately considering that this pathway is involved in retrieval stopping/suppression as suggested by Anderson and Floresco. We have attempted to make this more clear in introduction and discussion.

5) Minor: In Fig1.E, the authors can consider to color code the data points as that in the following panels (1F-1M), so that readers can easily connect the data presented in different figure panels. There are a few typos, and references planned but not added.

We have made these changes suggested by both you and Reviewer #1. Thank you for your attention to detail.

Reviewer #3 (Remarks to the Author):

The manuscript submitted by Totty and colleagues describes a series of experiments aimed at investigating the role of the nucleus reuniens in mediating: 1) synchronized activity between the prefrontal cortex and hippocampus, and 2) extinction-induced suppression of fear behavior. Previous studies have already established a role for the nucleus reuniens in both prefrontal-hippocampal synchrony and in fear suppression, but no previous study has directly connected these 2 functions of the nucleus reuniens. The study by Totty and colleagues for the first time directly connects these 2 functions by manipulating the nucleus reuniens in rats subjected to tone fear conditioning and extinction while recording both prefrontal-hippocampal synchrony and fear behavior. They found that within the same behavioral paradigm inactivating the nucleus reuniens causes a decrease in prefrontal-hippocampal synchrony and a decrease in extinction-induced fear suppression, while stimulation of the nucleus reuniens at the same frequency as the observed prefrontal-hippocampal synchrony promotes fear suppression. Combined, the results reported by Totty and colleagues provide an important contribution to the mechanistic understanding of extinction-induced fear suppression and to possible translational approaches for treating disorders associated with excessive fear.

The experiments, results, and conclusions are very clearly described. I do have some questions and suggestions.

1) The data shown in figures 4-6 were obtained using a counter-balanced design, with the data collapsed in the figures. Does this mean that no test-order effects were observed? It would be helpful to add supplemental figures where the data are not collapsed, so that readers can determine if there could be any interesting trends towards test-order effects.

Thank you for pointing this out. Indeed, there were no test-order effects observed throughout these studies. We have included a supplemental figure (SFig. 7) showing the uncollapsed data and statistics for the experiments in figures 4-6.

Supplemental Figure 7: Within-subjects behavioral data from Figures 4, 5, and 6 plotted by test order. There were no test-order effects in any of the within-subjects behavioral tests in either Fig. 4 (main effect of Test Order: $F_{1,9} = 0.171, p = .688$), Fig 5 (main effect of Test Order: $F_{5,25} = 1.208, p = .3342$), or Fig 6 (main effect of Test Order: $F_{1,18} = 0.002, p = .968$). This is visually shown by the similar line slopes in animals of the same group regardless of test order (e.g., On > Off or Off > On). All data are means \pm s.e.m.s.

2) In figure 6 the effects of 8Hz stimulation of the nucleus reuniens are reported. In panel 6H the effects are compared between ChR2- and mCherry-expressing rats. According to the legend this panel shows that “sinusoidal 8 Hz stimulation of RE reduces fear renewal in rats expressing ChR2 (n = 6) and not mCherry (n = 6)”. Visual inspection of the panel suggests that this description might not fully capture all the effects, as the absolute effect of light ON in the mCherry group seems larger than in the ChR2 group (though in opposite direction). Is there an effect in the mCherry group or not? In general, it would be helpful if the statistical results for all the data shown throughout the manuscript, whether significant or not, were added in a supplemental table. This would help readers understand how the conclusions presented throughout the manuscript were drawn.

See the question below and our response updating the sample size for this experiment. We have made sure to include post-hoc analyses for all data shown in this experiment to help the readers understand how the conclusions were drawn. We believe that the additional data and statistical tests now reported throughout the manuscript and supplemental document are sufficient for readers to understand how conclusions were drawn.

3) Also related to figure 6H, if there is an effect of light ON in the mCherry group, then what is the explanation of a light effect in rats that do not express an opsin? The answer seems relevant to interpreting the effect in the ChR2 group, as the light effect in the ChR2 group seems relatively minor and appears to somehow negate the light effect in the mCherry group.

As an addition to the experiment requested by Reviewer #2, we have replicated and increased the N for the experiment reported in figure 6. With this increased sample size, there is now a clear effect that ChR2 rats do indeed show reduced renewal under 8 Hz stimulation. However, as this reviewer points out, there is also a clear effect where the 8 Hz stimulation is actually increasing the freezing in mCherry control rats. We speculate that this is likely due to external disinhibition (PMID: 23791555); a phenomenon by which an unexpected, novel cue (in this case the blue light that escapes the patch cord/head cap and is visible to the rat) presented alongside an extinguished CS drives fear relapse. Importantly, both the recovery of extinguished fear via context-induced renewal and external disinhibition are thought to be driven by similar mechanisms relying on the hippocampus (see the above reference), making it plausible that this effect was blocked by RE activation. Thus, in the current manuscript, we speculate that the perceptible laser stimulus likely drove enhanced renewal in mCherry rats, and this effect was blocked by RE stimulation. In support of this argument, we show that 1) freezing in mCherry-expressing rats was specific to the CS and not the laser (**SFig 6**), and 2) the effect was robust no matter the test order (**SFig 7**). Although exhaustively testing this behavioral phenomenon is outside the scope of this manuscript, we have included all of these data and accompanying statistics to supplemental figures.

“We speculate that the moderate increase in freezing observed in mCherry rats is likely due to external disinhibition driven by the visually perceptible laser stimulus (Maren 2014), effectively driving further renewal. This effect was attenuated by RE stimulation in ChR2-expressing rats. Importantly, optogenetic stimulation prior to CS presentations had no effect on freezing behavior (**SFig 6**), and there were no test-order effects for any of the within-subject experiments reported in this paper (**SFig. 7**).”

4) In figure 1N-P, coherence in the 6-9Hz range is shown. It would be helpful to add in a supplement the 3-

6Hz range coherence data.

This is now shown in supplemental figure 1 (SFig 1A; see below).

5) Similarly, in figure 4F the coherence is only shown in the 6-9Hz range, and it would be helpful to add the 3-6Hz range data in a supplement.

This is now shown in supplemental figure1 (SFig 1B).

3-6 Hz Coherence

Supplemental Figure 2: PFC-HPC coherence in the 3-6 Hz frequency range. (A) Related to Figure 1, although 3-6 Hz peak coherence does appear to increase across test sessions (main effect of Day: $F_{2, 20} = 4.88, p = .019$), Tukey's post hoc comparisons did not reveal any significant effects (all $p > .06$). (B) Related to Figure 4, 3-6 Hz coherence was not affected by muscimol inactivation of the RE (main effect of Drug: $F_{1, 20} = 0.305, p = .587$).

6) The optogenetic manipulations reported in figures 5-6 were initiated before the CS onset instead of at the CS onset, even though figure 2 shows that theta-range rhythms in the nucleus reuniens start at CS onset. It would be helpful if a rationale for this difference in timing is included, and also to include the freezing data from the time-interval preceding the onset of the first CS while the light was already turned on.

Supplemental Figure 3: The evolution of freezing behavior during the first CS trial of Retrieval testing shown in Figure 4. (A,C,E) Two-second bins of freezing behavior were averaged starting from 20 seconds preceding the first CS onset to the 10 second period after CS onset for Jaws, GFP, and control (CTRL) groups during both laser ON and OFF conditions. Red laser illumination began 10 seconds prior to CS presentation. To determine if the laser had any effect before CS trials began, the freezing during this 10 second pre-CS period was averaged for each group. Although there was no significant Laser x Group interaction ($F_{2, 27} = 2.379, p = .112$), planned comparisons using Fisher's LSD show that red laser stimulation increased freezing in the Jaws group prior to CS presentations ($p = .027$), but not the GFP ($p = .426$) or CTRL groups ($p = .599$). All data are means \pm s.e.m.s; * $p < 0.05$.

Supplemental Figure 6: The evolution of freezing behavior during the first CS trial of Renewal testing shown in Figure 6. (A) Time course of freezing behavior in ChR2 and mCherry-expressing rats during both laser ON and OFF test sessions in relation to Laser and CS duration. (B) Laser presentation did not affect freezing behavior during the 5 seconds prior to the CS (Pre-CS period; Laser x Group interaction: $F_{1, 21} = 1.26, p = .275$). During the CS (C), mCherry rats show increased freezing when the Laser was turned ON (CS period; Laser x Group interaction: $F_{1, 21} = 5.46, p = .029$). Comparatively, (D) ChR2-expressing rats showed a quick reduction in fear following CS and laser termination (Post-CS period; Laser x Group interaction: $F_{1, 21} = 10.44, p = .004$). All data are means \pm s.e.m.s; * $p < 0.05$; ** $p < 0.01$.

REVIEWERS' COMMENTS

Reviewer #1 (Remarks to the Author):

All of my concerns have been adequately addressed in the revised manuscript.

Reviewer #3 (Remarks to the Author):

The authors have addressed all my questions and suggestions.